

# Spatially-distributed tracer-aided runoff modelling and dynamics of storage and water ages in a permafrost-influenced catchment

Thea I. Piovano[1], Doerthe Tetzlaff[2,3,1], Sean K. Carey[4], Nadine J. Shatilla[4], Aaron Smith[3], Chris Soulsby[1]

[1]Northern Rivers Institute, School of Geosciences, University of Aberdeen, AB24 3UF, UK
[2]Department of Geography, Humboldt University Berlin, 12489 Berlin, Germany
[3]IGB Leibniz Institute of Freshwater Ecology and Inland Fisheries, 12587 Berlin, Germany
[4]School of Geography and Earth Sciences, McMaster University, Hamilton, L8S 4K1, Ontario

*Correspondence to*: Doerthe Tetzlaff (d.tetzlaff@igb-berlin.de)

**Abstract.** Permafrost strongly controls hydrological processes in cold regions, and our understanding of how changes in seasonal and perennial frozen ground disposition and linked storage dynamics affects runoff generation processes remains limited. Storage dynamics and water redistribution are influenced by the seasonal variability and spatial heterogeneity of frozen ground, snow accumulation and melt. Stable isotopes provide a potentially useful technique to quantify the dynamics of water sources, flow paths and ages; yet few studies have employed isotope data in permafrost-influenced catchments. Here, we applied the conceptual model STARR (Spatially distributed Tracer-Aided Rainfall-Runoff model), which facilitates fully distributed simulations of hydrological storage dynamics and runoff processes, isotopic composition and water ages. We adapted this model to a subarctic catchment in Yukon Territory, Canada, with a time-variable implementation of field capacity to include the influence of thaw dynamics. A multi-criteria calibration based on stream flow, snow water equivalent and isotopes was applied to three years of data. The integration of isotope data in the spatially distributed model provided the basis to quantify spatio-temporal dynamics of water storage and ages, emphasizing the importance of thaw layer dynamics in mixing and damping the melt signal. By using the model conceptualisation of spatially and temporally variant storage, this study demonstrates the ability of tracer-aided modelling to capture thaw layer dynamics that cause mixing and damping of the isotopic melt signal.

## 1 Introduction

High latitude regions are experiencing some of the most rapid rates of environmental change as a consequence of global warming (Adam et al., 2009; DeBeer et al., 2016; Walvoord and Kurylyk, 2016), with limited process-based benchmarks against which to assess the implications. Permafrost, seasonal frost and snowmelt strongly control the hydrology and mechanisms of runoff generation in subarctic and arctic regions (Woo, 2012). There is an enhanced interest in understanding runoff processes in permafrost areas as they are highly susceptible to climate warming (Wright et al., 2008; Quinton et al.,



2009; Quinton and Baltzer, 2013; Endalamaw et al., 2017; Connon et al., 2018; Quinton et al., 2018). Knowledge of streamflow generation mechanisms and associated hydrological pathways is fundamental to understanding the functioning of these catchments, and the likely impacts of changes in energy and water availability (Quinton and Carey, 2008). However, despite the enhanced interest, logistical difficulties in gauging and monitoring remote northern environments limit empirical

studies and process understanding (Tetzlaff et al., 2015; Laudon et al., 2017). Thus, understanding how changes in permafrost distribution and associated storage dynamics affect the hydrology of catchments with permafrost is essential to reduce uncertainties in predicting the effects of climate change (Tetzlaff et al., 2018).

Permafrost depth and distribution are highly variable across circumpolar regions and exert a primary influence on runoff pathways, acting as an aquitard that restricts deep drainage and hydrological exchanges (Woo, 2012). Environments where

permafrost is spatially discontinuous are particularly complex as soil storage and water redistribution are also affected by the seasonal variability of the active layer (Carey and Woo, 1999; Quinton and Baltzer, 2013; Connon et al., 2018). Seasonally frozen ground (the active layer) acts as a transient subsurface zone that reduces permeability, limiting snowmelt infiltration and redistribution of water within the soils (Gray et al., 2001; Zhang et al., 2011). The influence of frozen ground on soil hydrology varies widely based on the depth of the frozen layer, the pore-size distribution and pore space volume containing

ice, and the temperature of the soils, which can result in considerable exchanges of latent energy (Walvoord and Kurylyk, 2016). While the physics of heat and mass transfer are well defined, our ability to accurately observe and simulate these processes is complicated by the inherent variability in soil properties and the lack of instrumentation to track all phases of water in the soil (Zhang et al., 2011). Furthermore, in the Subarctic and Low Arctic, the widespread presence of highly permeable organic soils influences runoff generation mechanisms (Quinton and Marsh, 1999; Carey and Woo, 2001a). The

occurrence of overland flow is typically limited, with subsurface drainage across hillslopes in the near-surface organic layer acting as the primary flow pathway, particularly during freshet and large rain events (Dingman, 1971; Quinton and Marsh, 1999; Quinton et al., 2000; Carey and Woo, 2001b).

Unlike rainfall-dominated catchments, the heterogeneous pattern of snow accumulation and melt introduces complexities when inferring the hydrological response of cold catchments. Vegetation and topography act as first-order controls on snow

accumulation and redistribution as  wind-trapping from shrub vegetation causes increased accumulation and redistributed snow settles in areas of reduced wind velocity (Pomeroy et al., 2006; Bewley et al., 2007; MacDonald et al., 2009). The widespread expansion of shrub vegetation is a notable feature throughout northern circumpolar regions (Tape et al., 2006) due to a warming climate, which influences surface-atmosphere exchanges and melt (Liston and Sturm, 2002; Marsh et al., 2010; Ménard et al., 2014). Topography has a similarly important role through slope and aspect effects, with the energy

balance of northern catchments resulting in runoff contributing areas that are highly variable in space and time (Quinton and Carey, 2008).

Logistical difficulties associated with access and data collection in many high latitude catchments limit the possibilities for empirical studies and process understanding. This makes stable isotopes a potentially useful tool for hydrological monitoring and to advance our understanding of processes. Stable isotopes are a useful method to estimate water transit or travel times



(TTs), defined as the elapsed time between water entry to, and exit from, a catchment as stream discharge at the outlet. Very few studies have employed isotope methods in permafrost influenced catchments to determine source waters (McNamara et al., 1997; Metcalfe and Buttle, 2001; Hayashi et al., 2004; Carey et al., 2013b; Tetzlaff et al., 2018). Often, isotopic data were employed to explore runoff processes based on hydrograph separation techniques or conceptual models for individual

events (Quinton et al., 2005; Boucher and Carey, 2010; Lessels et al., 2015), but to our knowledge no spatially distributed tracer-aided models have been applied to snowmelt-dominated permafrost catchments.

Estimating TTs and water ages within a catchment requires models that incorporate time-variant input signals from snowmelt and soil thaw that can be difficult to measure in cold heterogeneous environments. In addition, the various time-variant restrictions on infiltration during melt and on percolation as thaw progresses result in variable flow pathways that

influence TTs in a complex manner (Walvoord and Kurylyk, 2016). Capturing these processes in hydrological models remains a challenge. The STARR (Spatially distributed Tracer-Aided Rainfall-Runoff) model facilitates fully distributed simulations of hydrological storage dynamics and runoff processes, as well as the associated isotopic compositions and age distributions (van Huijgevoort et al., 2016). The most recent advancements in STARR improve modelling of spatially distributed mass balances of snow accumulation and melt along with simulation of the isotopic composition of melt water

(Ala-aho et al., 2017a, 2017b). STARR is now capable of simulating interactions between water storage, flux and isotope dynamics with multi-criteria calibration in snow-influenced environments (Piovano et al., 2018).

The Wolf Creek Research Basin (WCRB), together with an alpine sub-catchment (Granger Basin, GB), is a long-term experimental field site in subarctic Canada with various gauging and weather stations and a rich available data record (Rasouli et al., 2019). WCRB has been widely used as an international inter-comparison site for hydrological and

biogeochemical processes research and model development (Pomeroy et al., 2008; Rutter et al., 2009; Carey et al., 2010, 2013a; Laudon et al., 2013) and is considered a representative montane discontinuous permafrost catchment (Janowicz et al., 2004).

In this paper, we further develop the STARR model to simulate the spatio-temporal dynamics of storage in the permafrost influenced GB alpine catchment of WCRB. The specific objectives are to:

(i) Adapt STARR for discontinuous permafrost catchments to capture thaw layer dynamics that vary in time and space.

(ii) Conduct a multi-criteria calibration to simulate isotope fluxes in snowpack dynamics and streamflow.

(ii) Use the model to assess the temporally and spatially variant storage dynamics in soils influenced by freeze-thaw processes and the associated ages of resulting water fluxes.

This study demonstrates the value of tracer-aided modelling to quantify catchment water storage and age dynamics in

permafrost influenced environment for the first time.



## 2 Study site and data

### 2.1 Study site

The 7.8 km$^2$ Granger Basin (60°32′ N, 135°11′ W) is a subarctic alpine catchment within the Wolf Creek Research Basin (WCRB) (Fig. 1a), 15 km south of Whitehorse, Yukon Territory, Canada. WCRB has a dry seasonal subarctic climate

(Koppen classification *Dfc*) with a 30-year climate normal (1981-2000) reported for Whitehorse Airport (706 m a.s.l) of -0.1ºC and an annual precipitation of 262.3 mm, with approximately 40% falling as snow. However, precipitation at WCRB ranges in magnitude and phase with elevation (Rasouli et al., 2019), and Granger Basin (GB) on average has considerably higher precipitation than Whitehorse.

The elevation of Granger Basin ranges between 1310 and 2080 m a.s.l. (Fig. 1b). At lower elevations, the main river valley

trends west to east with predominantly south and north facing slopes (Fig. 1c, d). The geology is primarily sedimentary, consisting of limestone, siltstone, sandstone and conglomerate, overlain by a mantle of glacial till (Carey et al., 2013a). Atop bedrock, stony till and other glacial drift covers most of the basin. Soils in the top metre below 1650 m a.s.l. are sandy to silty and at lower elevations, the upper layer of soil is an organic layer with variable thickness up to 0.4 m consisting of moss, lichens and peat (Fig. 1f). The organic layer has an average porosity $\varphi = 0.88$ and an average density $\rho = 124$ kg m$^{-3}$,

while the deeper mineral layer has an average porosity $\varphi = 0.49$ and an average density $\rho = 1104$ kg m$^{-3}$ (Quinton et al., 2005).

Approximately, 70–80% of the Granger Basin is assumed to be underlain by permafrost (Lewkowicz and Ednie, 2004), whose disposition is controlled primarily by elevation and aspect. North-facing slopes and higher elevation areas are considered to have permafrost, whereas south-facing slopes are dominated by seasonal frost (Carey, 2003). The basin is

20 situated in the shrub taiga and alpine tundra ecozones, above treeline (~1200 m a.s.l.). Vegetation includes low-lying grasses, willow shrubs (*Salix Sp.*) and Birch (*Betula Sp.*) at lower elevations. Only few white spruces (*Picea glauca*) are present and tall shrubs (> 2 m) are present along the riparian corridor in the lower basin (Fig. 1e). The upper portion of the basin is dominated by bare rock and alpine tundra with limited vegetation (Quinton et al., 2004; McCartney et al., 2006; Carey et al., 2013a).

### 2.2 Data

Meteorological data (air temperature, wind speed, solar radiation, relative humidity) were recorded every 15 minutes at Buckbrush weather station (BB, 1312 m a.s.l.), located ~2.8 km from the Granger Basin outlet in an area of WCRB with similar characteristics to Granger Basin (Fig. 1a). To fill gaps in meteorological data, a linear regression between the Alpine weather station (1615 m a.s.l.) 6.2 km from BB within WCRB was used over the period 2008-2017 (with a R$^2$ on average

>0.67; air temperature had the best fit with a R$^2 \sim 0.96$). Air temperature records were adjusted for altitude effects, assuming the moist adiabatic lapse rate of -0.006°C m$^{-1}$ a.s.l. (Goody and Yung, 1995). The daily precipitation time series integrated data from two precipitation gauge instruments at the BB weather station: a Geonor T-200B gauge deployed in a standard





configuration with a single Alter-shield and a tipping bucket rain gauge (TBRG). The challenge of measuring precipitation in these regions has been shown in a comparison of several catchments conducted by Pan et al. (2016), which included BB. According to the experimental relationship suggested for Wolf Creek, a wind factor correction has been applied to the precipitation. Finally, altitude effects were considered, assuming a 0.05% increase in precipitation for each 100 m increase in

elevation. Rasouli et al. (2019) describe in detail the elevation-dependent climatology of WCRB and the relationship between meteorological variables among the long-term weather stations used in this study.

Automated snow pillow measurements from the BB weather station were averaged to obtain a daily time series of snow water equivalent (SWE). The instrument converts the weight on the snow overtop the snow pillow into SWE. Stream discharge was calculated at the outlet of Granger Basin using a rating curve updated each year. A stilling well was

10 instrumented with a pressure transducer (Solinst Levelogger) and compensated with a co-located Solinst Barologger measuring pressure/stage every 15 minutes. Manual flows were taken frequently using a SonTek Flowtracker for a range of flow conditions and with salt dilution gauging during periods when the channels were ice-covered. The loggers were in place from mid-April to October when they were removed to prevent freezing.

Stable water isotope samples ($^2$H) were collected from stream water, precipitation and snowmelt. Stream water isotopic

composition was taken both from grab samples and an ISCO autosampler located at the gauging station of GB. The samples had an average sampling frequency of <2 days both in 2015 and 2016 during the sampling period (mid-April to October, same as the discharge measuring period). Only a few samples were collected with snowmelt lysimeters in 2015 and 2016 at different locations in the bottom valley of Granger Basin. Rainfall samples were collected using samplers adapted after Gröning et al. (2012) from different locations both in Granger Basin and WCRB. Stable isotope ratios of hydrogen and

oxygen were determined using a Los Gatos Research DTL-100 Water Isotope Analyzer at the University of Toronto. Five standards of known isotope composition, with δ$^2$H ranging from −154‰ to −4‰, purchased from Los Gatos Research were used for calibration, in addition to periodic checks using the international standard VSMOW2. During analytical runs, samples were interweaved with standards at a ratio of 3:1. For periods when precipitation isotopes were not sampled, they were estimated via a linear regression between precipitation δ$^2$H and air temperature. In addition to temperature and

precipitation time series, the precipitation isotopic composition was spatially distributed to account for altitude effects at a rate of -0.04‰ m$^{-1}$ a.s.l. for δ$^2$H (Holdsworth et al., 1991).

Soil hydrological properties were mapped by splitting the model domain (GB) into two main hydropedological units, each with different properties and hydrological responses: the upper basin (UB) and lower basin (LB) (Fig. 1f). The first unit (UB) includes areas at upper elevations (>1650 m a.s.l.) where bare soil is classified as *regolith*, bedrock outcrops occur and

30 the organic layer is absent. The second unit (LB) aggregates the other soil classes comprised of organic soil of varying thickness including riparian soil. The model domain was divided into 4 classes based on vegetation properties (Fig. 1e): a *tall shrub* class that occurs mainly in the riparian areas of the lower basin, shrub at lower elevations but not in riparian areas, short shrub in areas in the central basin and *no vegetation* in the upper basin.



**3 Model development for frozen soil and thaw layer dynamics**

The Spatially distributed Tracer-Aided Rainfall-Runoff (STARR) model is a spatially explicit hydrological model aimed at simulating water fluxes, storage dynamics, isotope ratios, and water ages. It is based on a HBV-type conceptual model structure (Lindström et al., 1997) for representing soil and groundwater storages and fluxes both in terms of water flow and

isotopic composition. Briefly, the model equations of the different routines (interception, snow, soil and groundwater) are applied to each grid cell. For each compartment, isotope ratios are estimated according to mixing equations with the assumption of complete mixing. The snow module is built on the assumption that the interception efficiency decreases with canopy snow load and increases with canopy density; and further that snow unloading increases with time (Ala-aho et al., 2017a). The snow module is also energy-based, hence it accounts the sublimation fractionation of snow isotopes both of

canopy intercepted snow and the ground-level snowpack and the depletion of isotopic composition of snowmelt. The water ages are estimated by tracking the storage cell by cell at each time step, so the water ages dynamically and spatially evolve. Full details about the model structure, parameters and equations are given in (van Huijgevoort et al., 2016; Ala-aho et al., 2017b). In this application, according to the parameter sensitivity analysis conducted in the previous applications of the model, some of the parameters were randomised from initial ranges and then calibrated to optimise the model in a Monte

Carlo approach (Table 1). Other parameters were kept fixed to specific values selected through preliminary testing runs.

The overarching goal of STARR is to keep the model structure simple while representing the spatial distribution of key hydrological processes for a given environment; in this case, a subarctic alpine environment with discontinuous permafrost. Applying the model to GB required conceptualisation of the impact of frozen soil by modelling the field capacity of the soil as time-variant to represent active layer storage (Carey and Woo, 2001b). The approach used was to i) limit available soil

storage when the soil is frozen, ii) make available soil storage gradually increase when temperatures are above 0ºC to a maximum value, and then iii) reduce available soil storage during the refreezing period (Fig. 2).

The freezing dynamics were implemented by setting the field capacity $FC$ to a minimum value ($FC_{min}$) when $t < t_1$ (where $t_1$ is the beginning of the thaw period) or $t > t_4$ (where $t_4$ is the end of freeze back) linearly increasing for $t$ between $t_1$ and $t_2$ (where $t_2$ is the time of maximum thaw) up to the maximum value $FC_{max}$ and linearly decreasing between $t_3$ (where $t_3$ is

the onset of freeze back) and $t_4$ (Eq. 1, Fig. 2).

$$FC(t) = FC\,(t-1) + \Delta FC \qquad (1)$$

$$\begin{cases} FC(t_0) = FC_{min} \\ \Delta FC = 0 & t < t_1, t > t_4, t_2 < t \le t_3 \\ \Delta FC = (FC_{max} - FC_{min}) \cdot (t_2 - t_1)^{-1} & t_1 \le t \le t_2 \\ \Delta FC = (FC_{max} - FC_{min}) \cdot (t_3 - t_4)^{-1} & t_3 \le t < t_4 \end{cases}$$

In addition, to make this frozen soil approximation spatially consistent with field observations, a parameter ($ASP$) accounting for aspect was included. Specifically, south-facing slopes were considered to have consistently higher available soil storage

than the cells facing different cardinal directions. The parameter $ASP$ was set to 1 if the cell was non-south facing and randomly sampled from the range [1.0-1.2] for south facing cells. This indicates that the time variant available storage for a





south facing cell was permitted to be at maximum 20% higher than a non-south facing cell. The *FC* is the product of soil depth (*sd*) and the volumetric field capacity (*fcap*), the time variant *FC* is defined from $FC_{min}$ (Eq. 2) and $FC_{max}$ (Eq. 3):

$$FC_{min} = sd \cdot fcap \text{ (2)}$$

$$FC_{max} = FC_{min} \cdot fcr \cdot ASP \text{ (3)}$$

Where *fcr* is the field capacity rate parameter used to calibrate the ratio between $FC_{min}$ and $FC_{max}$. To avoid unreasonable *FC* values, the *fcap* parameter was randomly sampled and optimised during the calibration process. For simplicity, the intervals of the piecewise function (Eq. 1) were set *a priori*, according to site knowledge based on previous observations of frozen ground development (Zhang et al., 2010). The endpoints were set as following: $t_1 = 1$ May, $t_2 = 15$ July, $t_3 = 1$ October and $t_4 = 1$ November. Thus, *FC* was set to its minimum for November-April, linearly increasing from May to mid-

July, at maximum until October and finally sharply decreasing to represent soil freeze back. Despite most parameters values being set for the whole catchment, some were set to reflect the properties related to soil, vegetation and aspect classifications (Fig. 3).

A multi-variate calibration approach (Ala-aho et al., 2017b; Piovano et al., 2018) was applied to optimise the model according to three variables: discharge, SWE and stream water isotope values. To obtain model efficiencies of discharge and

stream water isotopes, we compared observations of both at the gauging station with simulated values at the outlet cell. For SWE, we compared the observed SWE at BB and the catchment average of simulated snow pack for each time step. However, the simulated values were dependent on elevation, and it was not possible to compare simulations and observations directly as measurements were taken at BB, outside of Granger Basin.

We ran 7000 simulations in a Monte Carlo approach, each one characterised by a parameter set randomly sampled from an

initial range (Table 1). In each simulation, years 2014 and 2015 were looped in order to initialize the system in terms of stabilizing both water storage and isotope values. A multi-variate calibration was then conducted for the period 2014-2016 and for each simulation three efficiencies were determined: Kling-Gupta Efficiency KGE (Gupta et al., 2009) for evaluating discharge and SWE and Mean Absolute Error MAE for evaluating $\delta^2$H simulations. The empirical cumulative distribution functions of these efficiencies were used to select the 100 simulations showing simultaneously the highest values of KGE for

discharge and SWE, and the lowest value of MAE for stream water $\delta^2$H. KGE was chosen for hydrometric variables as it is based on equal weighting of linear correlation, bias ratio, and variability. KGE tends to be a more balanced approach than the Nash-Sutcliffe NSE (Nash and Sutcliffe, 1970) with less biasing to peak runoff (Gupta et al., 2009). For evaluating simulation results for stable isotopes, MAE was chosen as the error range estimates are in the same scale as observation variability and MAE provides advantages over the root mean square error (i.e. MAE is a more stable measure of the

magnitude of average error, while the RMSE can be confounded by the variable functional relationship between RMSE and average error (Willmott and Matsuura, 2005)). Time series results were based on the 100 best simulations, while the spatial model outputs were based on the best simulation according to the same method used for the 100 best simulations, and therefore were dependent on the intrinsic characteristics of the selected simulation.





The approach used in STARR allows the estimation of the spatial distribution and dynamic variation of water ages. In addition to the isotopic composition, water ages were computed according to a full-mixing assumption within each cell and time step according to passive storage parameters for soil and groundwater reservoirs in the model (Table 1). Water ages of total runoff (i.e. overland flow, and lateral soil and groundwater fluxes) were estimated using a mass balance for each cell

that tracked all inflows and outflows, and quantified total runoff. During each time step, water stored in each model compartment becomes a day older, with the age of incoming precipitation taken as 1 day. The age of water in each compartment evolves dynamically through mixing and water exchange between compartments and cells. Stream water ages evaluated at the outlet cell are the time-variant integration of all water fluxes from different catchment compartments, each having their own age distributions (similar to Soulsby et al., 2015). All simulations were performed at 100 m x 100 m

resolution and at a daily time step. The study period is 1 January 2014 to 31 December 2016.

## 4 Results

### 4.1 Temporal dynamics of water fluxes, storage and age

During the study period, measured discharge ranged from 0.001 $m^3s^{-1}$ to 1.1 $m^3s^{-1}$ (Fig 4). There were limited discharge measurements between October and April, except for occasional under-ice measurements. Seasonal variability in flow was

more marked than the inter-annual variation (standard deviation across the sampling period (approximately April-October for all analysed years): $\sigma$ = 0.139, while across the sampling period (again April-October) of each year: $\sigma_{2014}$ = 0.147, $\sigma_{2015}$ = 0.145, $\sigma_{2016}$ = 0.116). In each year, the first recorded flow peak was related to snowmelt, while later in the summer discharge was most responsive to large rainfall events (Fig 4a, b). In 2014, the annual peak discharge peak (1.1 $m^3s^{-1}$) occurred at the beginning of July following a 36 mm rain event, whereas in 2015 the annual peak was 0.67 $m^3s^{-1}$ recorded at the end of May

during the spring melt. In 2016, peak flow was (0.85 $m^3s^{-1}$) was measured in mid-June following a 31 mm rain event, at the end of the snowmelt period.

Measured precipitation $\delta^2H$ ranged from -76.9‰ to -180.2‰, showing some seasonality with more depleted values in winter and more enriched in summer. In contrast to the variable precipitation signal, the stream water $\delta^2H$ measurements exhibited a highly damped signal that lacked the clear depression usually associated with alpine catchments during the snowmelt. If

anything, $\delta^2H$ shows some enrichment from snowmelt period until the summer, albeit with some limited short-term variability (Fig. 4c). The sparse snowmelt $\delta^2H$ samples were on average -155.3‰, which is higher than the minimum stream water values. The snowpack generally begins to accumulate in late August/early September, reaching a peak the following spring before the main melt period in late April to May. The highest value of observed snow water equivalent SWE (230 mm) was measured in April 2014 (Fig. 4d). In 2016, peak SWE was lower than in the other years (112 mm).

Given the complexity of the catchment hydrology, the model simulated discharge dynamics quite well over the three years, with KGE values ranging between 0.70 and 0.48 across the 100 best simulations (Fig. 4b, Table 2). The snowmelt freshet was well reproduced both in terms of timing and peak, yet the simulated recession limbs were slightly steeper than observed.





The early summer period was characterized by the highest discharge peaks both in 2014 and 2016, which was well represented by the model. The largest underestimation of the simulated discharge was during July and August 2016 during rain-driven events. Overall, the median absolute difference between daily observed discharge and median simulation was larger in 2016 (0.12 $m^3$ $s^{-1}$) than 2014 (0.06 $m^3$ $s^{-1}$) and 2015 (0.05 $m^3$ $s^{-1}$). The main challenge in reproducing the stream

water $\delta^2H$ composition (Fig. 4c) was the limited seasonal variability of the stream water signal, coupled with some shorter day-to-day variation. Despite the difficulties in reproducing this damped signal, the span of the efficiency across the 100 best runs (MAE between 4.41 and 5.84 ‰) was similar to the model performance obtained in previous model applications in snowmelt-driven catchments (Ala-aho et al., 2017b; Piovano et al., 2018). The simulated signal did reproduce a slight depletion after melt followed by enrichment, although the measured melt signal showed greater seasonal mixing and

damping than the simulated, though greater day-to-day variation.

Observed and simulated (averaged across the catchment) temporal SWE dynamics are shown in Fig. 4d. The snowpack model output was highly dependent on elevation, yet the spatially-averaged time series reasonably reproduced the SWE measurements with performance efficiencies of KGE ranging between 0.71 and 0.72 across the 100 best simulations. Overall, catchment average SWE was underestimated in 2014, and overestimated in 2016 despite capturing the peak. In

2015, the first part of the accumulation period was overestimated by the model, whereas the second part underestimated accumulation prior to melt. The melt period in 2015 was shorter than in 2016 and the decline from high values of SWE to 0 more rapid (26 days in 2015 from early May to end of May, against 61 days in 2016 from mid-April to mid-June). Moreover, in early April 2016 the simulated melt period was interrupt by a period of refreezing, which delayed the simulated melt compared to that measured. Due to the scarcity of snowmelt isotope samples, it was not possible to include these

observations in the calibration (Fig. 4d). However, simulated snowmelt $\delta^2H$ was reasonable as the majority of the observations in 2015 and 2016 fell within the uncertainty bands of the simulated composition.

Fig. 4e shows the simulated stream water ages at the outlet cell of the catchment. The median water age over the calibration period was 290 days. Stream water ages showed clear seasonal dynamics: a steady increase to >1 year (with large uncertainty) when deeper sources sustained low flow periods in winter. Ages were reduced to <9 months in the early phases

of the melt, gradually decreasing to around 6 months in the late summer as younger water from melt and summer precipitation was an increasingly dominant source of runoff. The uncertainty was considerably reduced during high flow periods.

## 4.2 Spatial variability of water fluxes, storage and ages

Figures 5, 6 and 7 provide spatially distributed estimates of select model outputs (i.e. simulated SWE, soil storage volumes

and ages of water fluxes from each cell, respectively) at hydrologically relevant dates or averaged over specific periods. The patterns in the SWE maps (Fig. 5) reflect the influence of the vegetation parametrisation (e.g. presence of shrubs in the lower basin with interception and unloading processes) in addition to the influence of elevation. In 2015, the simulated snow accumulation was less variable across the catchment compared to the more heterogeneous snowpack in 2016. In 2015, melt



occurred with a short time lag between lower and upper elevations (16 days difference when SWE=0 in the lowest and highest catchment cells). In contrast, in early 2016 simulated SWE values were more heterogeneous across the catchment. Highest simulated values averaged across the catchment occurred when the lower catchment had values lower than 50 mm, whereas the upper basin reached 300 mm. In contrast to the rapid melt in 2015, the time lag between the day when SWE=0

occurred at the lowest cell and at the highest cell in the catchment was 59 days. This reflected the slower observed melt, though the modelled melt was delayed.

Spatially simulated liquid soil water storages are shown in Fig. 6 for different periods. The highest simulated values of soil water storage occurred immediately after the melt period in 2015 (Fig. 6a) and in 2016 (Fig. 6b). As the melt period was shorter and occurred more uniformly across the catchment in 2015 compared to 2016, the variability in soil storage across

the cells was lower in 2015 than in 2016 (standard deviations σ=13.8 mm compared with σ=22.2 mm in 2015 and 2016, respectively). In 2016, the upper basin cells received considerable water inputs (up to 183 mm) from melt, while in the lower basin the storage was lower than 36 mm. In addition, estimated soil water storage averaged on bi-weekly windows provided information on the spatial variability in soil water storage as the thaw layer increased while soils approached the highest overall field capacity (Fig. 6c and 6d). The highest field capacity occurred when the available storage was maximized (Fig.

6e and 6f) and then when available liquid storage was declining during freeze back (Fig. 6g and 6h). Overall, spatial patterns in the soil storage maps reflect a combination of the empirically-based parametrization of aspect (south facing and non-south facing) as well as the distinct vegetation and soil characteristics of the upper and lower basin.

Simulated ages of runoff (as combined overland flow and lateral flow from model cells) varied spatially across the catchment during the period simulated (Fig. 7), according to the spatial variation of storage reservoirs and water fluxes. Deep

groundwater contributions were generally negligible, consistent with the conceptual model of hydrology in GB, that account runoff generation occurring primarily from shallow lateral flow in the upper organic soil layers (Carey and Woo, 2001). Estimated water ages were predominantly influenced by soil runoff ages as overland flow and only occasionally simulated during melt. Overall, patterns of simulated water ages reflected flow path lengths and soil storage patterns, with ages increasing along drainage directions from hillslopes towards the valley bottom. Cells located at the convergence of the

longest flow paths had modelled water ages consistently older than in the other catchment cells (usually older than 300 days). From the oldest ages immediately following melt (catchment average of 295 days for 2015, Fig 7a and 233 days for 2016, Fig. 7b) as stored water was displaced, water ages decreased with increasing soil storage, summer precipitation and discharge in July (catchment average of 242 days for July 2015, Fig. 7c and catchment average of 234 days for July 2016, Fig. 7d), reaching the youngest overall ages at the end of the summer (catchment average of 177 days September 2015, Fig.

7e and 185 days for September 2016, Fig. 7f). Ages increased again in October (2015 in Fig. 7g and 2016 in 7h) when the availability of soil storage became restricted as refreezing occurred. Water ages after melt were greater in 2015 than in 2016. This is consistent with the different characteristics of the snowmelt periods in the two years, in term of timing, magnitude and heterogeneity across the catchment. As in 2015 melting was more homogenous compared to 2016 and the day after complete snowmelt (i.e. the first day with SWE=0 in all cells) occurred less than one month after highest values of SWE, the



hydrological system was still affected by the mobilization of old water in storage and the influence of newer melt water became stronger several days later. However, the melt period in 2016 was longer with a gradual decrease of the snowpack (two months from high values of SWE to 0) and so the model compartments had already time to be filled with newer water inputs which made the overall age pattern younger.

## 5 Discussion

### 5.1 How well can spatially distributed tracer-aided modelling capture thaw layer dynamics and runoff generation in permafrost catchments?

In discontinuous permafrost environments, storage dynamics and water distribution are strongly affected by the seasonal variability of frozen ground, the heterogeneity in snow accumulation and melt, and (particularly in alpine areas) the strong variability in energy receipt due to low sun angles (Quinton and Carey, 2008; Woo, 2012). There has been limited application of isotopes to investigate water sources and pathways in permafrost regions (McNamara et al., 1997; Metcalfe and Buttle, 2001; Hayashi et al., 2004; Carey et al., 2013b; Tetzlaff et al., 2018), and there have been fewer studies that have used hydrological models that explicitly incorporate tracers in permafrost and snow-dominated regions (e.g. Lessels et al., 2015). In this study, a spatially distributed model that integrates isotopic composition and discharge was applied to simulate runoff generation, stable isotopes, snowmelt and track water ages. By taking into account thaw dynamics, a time-variant available soil storage was implemented for the first time in the tracer-aided STARR model. Previous applications of STARR demonstrated that multiple data sets and calibration criteria helped constrain models and provided information on the relative value of observations (Ala-aho et al., 2017b; Piovano et al., 2018), which guide experimental design and provide information on gaps in process understanding (Kuppel et al., 2018). In this work, by incorporating a simple, but distributed conceptualization of thaw dynamics, STARR was able to effectively simulate the temporal dynamics of discharge and provide an adequate damping of the stream isotope signal. Despite the large variability storages and fluxes which are well documented for GB, the model was able to efficiently simulate the three calibration variables, providing confidence that the dominant governing processes are generally well simulated and parameters well represented. That said, although the overall damping of the isotopes is captured, the detailed short-term day-to-day variation is not. Also, the model exaggerates the effect of the depleted snowmelt pulse reaching the stream.

At WCRB and Granger Basin, previous research has documented how frozen ground status, organic soils, aspect and permafrost dominate the hydrological response to rain and snowmelt events (Quinton et al., 2004, 2005; McCartney et al., 2006). In complex alpine landscapes, aspect and altitude control the timing of melt and the subsequent delivery of water to soils. Organic soils are hydrologically complex, with a high porosity that can rapidly convey water to the stream when thawed or near 0ºC during melt. Lateral flow in organic soils is a key process in subarctic permafrost regions, as hydraulic conductivity is typically orders of magnitude greater than underlying mineral substrates (Carey and Woo, 1999, 2001b; Quinton et al., 2005). The descent of the frost table along permafrost slopes allows deeper flow pathways to become active



even as water tables typically fall, muting streamflow response (Carey et al., 2013a). Areas without permafrost are thought to contribute less to runoff, yet the overall disposition of permafrost in alpine landscapes is difficult to fully ascertain (Bonnaventure et al., 2012) and the relative influence of deeper groundwater flow is uncertain at present. Given this complexity, the day-to-day isotope dynamics are likely underpinned by more localised spatially heterogeneous processes

than modelled, which are probably not fully captured by the current model set-up (Ala-aho et al., 2017a).

This is because in this study, the STARR model was set up to best represent the process understanding described above by: (i) splitting the model domain into two hydropedological units based on organic soil presence, (ii) including a classification based on aspect to reflect permafrost presence and available soil storage, and (iii) allowing soil storage to be time-variant to reflect the seasonal development of frozen ground. Previous research in GB has highlighted the importance of spatially

representing processes in both field and modelling studies. For example, for the snowmelt period, McCartney et al. (2006) divided GB into nine hydrological response units (HRUs) based on similarities in hydrological, physiographic, vegetation and soil properties. Dornes et al. (2008) showed that a distributed hydrological model based on five HRUs was necessary to describe the observed magnitudes of both snowmelt and basin runoff. In this work, we explicitly model processes in GB at a 100 m scale, and model results support the observations that overland flow is largely absent in this basin (Carey and Woo,

2001a; Carey et al., 2013a) and that spatially represented melt is critical to accurately predicting snowmelt freshet (McCartney et al., 2006; Dornes et al., 2008). The spatial representation of processes in GB was also important in simulating biogeochemical processes as shown in Lessels et al. (2015). However, it is likely that the 100 m grid is too coarse to pick up small scale processes which might affect the day-to-day isotope variability.

**5.2 Spatio-temporal variations in water storage, flux and ages**

Spatial heterogeneity in modelled hydrological response was most strongly influenced by the timing and magnitude of ground freeze/thaw and also by the heterogeneities of inflows from snowmelt. Simulated melt rates were highly variable among years due to the disparate nature of snow accumulation and contrasts in melt rates across elevation, vegetation type and aspect. The differential melt rates affect various aspects of hydrological response including the distribution of soil

storage, isotopic composition and water ages. STARR explicitly tracks the spatial distribution of water isotopes from melt and rainfall inputs, and through the soil system to the stream. This approach provides an additional temporal dynamic of soil water ages throughout the watershed. Linking runoff generation processes to water ages is similar to, yet distinct from, hydrograph separation approaches that isolate sources of water (Laudon et al., 2002; Carey and Quinton, 2004; He et al., 2015). Other techniques for estimating water ages such as convolution integral transit time distributions (McGuire and

McDonnell, 2006), lumped conceptual models (Soulsby et al., 2015), and storage selection functions (Rinaldo et al., 2015) are useful, yet limited in terms of capacity to track the spatial distribution of water ages in the catchment. To our knowledge, there has been no estimate of spatially distributed water age in subarctic and arctic regions.



Simulated water ages at the end of the melt season were older than in any other season, as the main water source at this time is the stored water within the catchment which is being mixed with, and displaced by snowmelt (Fig. 4 and 7). In other words, the main source of water at the end of the snowmelt period is relatively old water that has been stored in the catchment over winter. The sources of water that lead to streamflow response have been previously studied in GB and other

permafrost catchments using stable isotopes and more traditional hydrograph separation techniques (Quinton et al., 2005; Boucher and Carey, 2010). Conceptual models often suggest that low storage capacities and high runoff ratios observed during the freshet must be dominated by inflow from directly connected melt, or 'new' water, as opposed to 'old' water that resides in the catchment over winter when soils are frozen. However, empirical results have been variable, with some catchments showing a predominance of new water (Cooper et al., 1991, 1993; McNamara et al., 1997; Hayashi et al., 2004)

or old water (Obradovic and Sklash, 1986; Gibson et al., 1993; Carey and Quinton, 2004; Carey et al., 2013a) depending upon site characteristics and/or the time of year. While the cause of the differing new and old water contributions is unclear, there appears to be considerable inter-annual variability based on hydrological conditions of the ice-rich soil and the nature of melt (Metcalfe and Buttle, 2001; Carey et al., 2013a). In this study, water ages as determined by STARR are oldest after melt and then decrease throughout the summer, and while there is considerable variability within the catchment, this pattern

generally holds until freeze back begins in October and soil storages decline. The decline in soil storage, however, represents a dynamic liquid water storage, not total water storage in soils. For GB, Carey et al. (2013a) reported event water contribution from two-component hydrograph separation ranging between 10 and 26%, suggesting that most of the water reporting to the outlet during freshet is old water (up to 90%). This observation agrees well with results of the STARR model, suggesting that streamflow water during freshet is primarily displaced old water that has resided in the watershed

over-winter.  This is also consistent with the findings of recent small catchment observations at Trail Valley in Canada (Tetzlaff et al., 2018) and larger scale observations in the watershed of the River Ob in Russia (Ala-aho et al., 2018b, 2018a). The mechanism of old water displacement from soil storage to the stream during melt results in the mixing and dampening of the isotope signal (Unnikrishna et al., 2002). This appears to be the case in GB, where despite the presence of frozen ground and permafrost, multiple years of study suggest that pre-event water predominates, and model results presented here

suggest water is oldest during freshet. The question remains as to how soils that are largely frozen or at 0°C during melt can contribute old water to the stream and that the new water signal from large volumes of melt water is strongly damped. The most plausible explanation is the presence of organic soils, which are able to mix and transmit water during melt, even at 0°C. The ability of organic rich soil horizons to mix and dampen isotope signals has been documented in other cold environments (Tetzlaff et al., 2014). In GB, organic soils cover much of the basin, and empirical studies have shown that

their high porosity allow several hundred mm of total water storage when saturated and large amounts at field capacity (Quinton et al., 2005). Depending upon their moisture content at freeze back, there may be considerable infiltration opportunity during melt (McCartney et al., 2006). Carey et al. (2013a) hypothesized a mechanism where in the early stages of melt, there is limited transfer of water to the stream until unsaturated storage capacity is satisfied and infiltrating melt water supplies sensible heat to bring soils to freezing (yet not necessarily thawed). At this time, mixing with pre-melt water



occurs and runoff generation begins. It is instructive that the contribution of old water increases throughout the melt period (Carey and Quinton, 2004; Boucher and Carey, 2010), which correspond to the age estimates from the STARR model. Furthermore, the differences in ages between years (Fig. 7), particularly during melt, support the observations that there is considerable inter-annual variability in the ages and source of water contributions, which are linked to conditions the

previous fall and the rate and timing of melt.

The use of STARR as a spatially distributed tracer-aided model provided a number of new insights into the hydrology of Granger Basin, which has a rich history of observation and model application. While prior work identified the dominance of 'old' water during freshet (Carey and Quinton, 2004; Carey et al., 2013a), the STARR model was able to support this finding in a conceptual model that captured the dominant runoff generation mechanisms and mixing processes. The role of organic

soils and variable soil storage was highlighted through multiple lines of evidence, and the importance of over-winter storage and melt dynamics on water ages was partially clarified. The implications of this enhanced understanding are important in terms of hydrological and biogeochemical response to climate change. In this region, temperatures are warming rapidly, particularly in winter (DeBeer et al., 2016). Changes in flows and biogeochemical fluxes for northern regions have been reported at larger scales (Frey and W., 2009; Tank et al., 2016), yet their link to process understanding at the headwater scale

remains scarce. The fact that there is considerable inter-annual variability in the hydrological and isotopic response suggests large potential variability in ages, which may amplify as the soil freezing regimes change. Furthermore, the influence of permafrost thaw on water ages remains unclear. Considering how water ages links to soil contact times and biogeochemical processes, further application of age estimation and understanding water storage dynamics may aid in understanding ecosystem response in a warmer world.

**6 Conclusion**

In this work, we applied the spatially-distributed, tracer aided model STARR to a cold, permafrost-influenced catchment in Yukon Territory, Canada. The model was subject to multi-criteria calibration with multi-year field data to simulate runoff, snowpack dynamics and stream isotope composition. The model captured all three variables reasonably well, and notably the highly damped stream isotope signature was reproduced. Critical to this was the conceptualization of the thaw layer

dynamics and thick organic soils, which provided a sufficient mixing volume that buffered the snowmelt isotope signature prior to water reporting to the stream network. Results from the model output correspond with previous field investigation and hydrograph separation studies that indicate relatively old water (pre-event) dominates runoff generation during spring freshet. The relatively flashy nature of spring freshet in this largely frozen alpine catchment may seem counter-intuitive to this finding, yet water stored within the catchment from the previous year is the main source of stream water at the end of the

melt season and explains isotopic damping of the signal. This result has important implications for processes that drive inter-annual variability and biogeochemical cycling as, there are considerable time lags in the system. How climate warming affects frozen ground and water ages is uncertain, yet water ages will likely increase with a reduction in cold. Another



important conclusion is that the snowmelt regime, which is variable each year, is an important factor in controlling water ages.

## Acknowledgments

This work was funded by the European Research Council (project GA 335910 VeWa). We also thank funding from the Natural Sciences and Engineering Research Council's Changing Cold Region Network and the Global Water Futures program. We gratefully acknowledge the assistance of Heather Bonn, Renée Lemmond, David Barrett and Tyler de Jong for their help in the collection of field data.

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

**Table 1: Parameters and their range of variation during calibration.**

| Category | Name | Unit of meas. | Description | Min | Max |
|---|---|---|---|---|---|
| SNOW | sfcorr | [-] | Correction factor for snowfall on top of wind correction | 0 | 0.3 |
| | ttlow | [°C] | Temperature below which all precipitation is snow | -2 | 0 |
| | tthigh | [°C] | Threshold temperature above which all precipitation is liquid | 0 | 2 |
| SOIL | fcap_ub | [-] | Volumetric field capacity (upper basin) | 0.3 | 0.8 |
| | fcr | [-] | Field capacity rate | 1 | 2.5 |
| | ASP (south facing cell) | [-] | Aspect field capacity rate (for south facing cells) | 1 | 1.2 |
| | ks_ub | [day-1] | Recession coefficient to determine outflow from soil storage (upper basin) | 5 | 40 |
| | ks_lb | [day-1] | Recession coefficient to determine outflow from soil storage (lower basin) | 5 | 40 |
| | KsPow | [-] | Power coefficient | 1 | 2 |
| | betaSeepage | [-] | Recession coefficient to determine soil recharge into groundwater | 0.1 | 1 |
| | LP | [-] | Fraction of limiting actual evaporation | 0.1 | 1 |
| | Cflux | [-] | Parameter for maximum capillary flux | 0.01 | 1 |
| ISOTOPES | deplOffset | [‰] | Offset parameter, equilibrium between ice and liquid | 0 | -19 |
| | Efrac | [‰] | Snow sublimation fraction | 0 | 80 |
| | GWpas | [mm] | Mixing volume groundwater | 1 | 200 |
| | SMpas | [mm] | Mixing volume soil | 1 | 200 |





**Table 2: Efficiency ranges of 100 best calibrations, selected to simultaneously have the highest KGE for discharge and SWE and lowest MAE for stream δ²H.**

| Variable | Max eff | Min eff |
|---|---|---|
| Q | KGE = 0.70 | KGE = 0.48 |
| SWE | KGE = 0.71 | KGE = 0.72 |
| Stream δ²H | MAE = 4.41‰ | MAE = 5.84‰ |

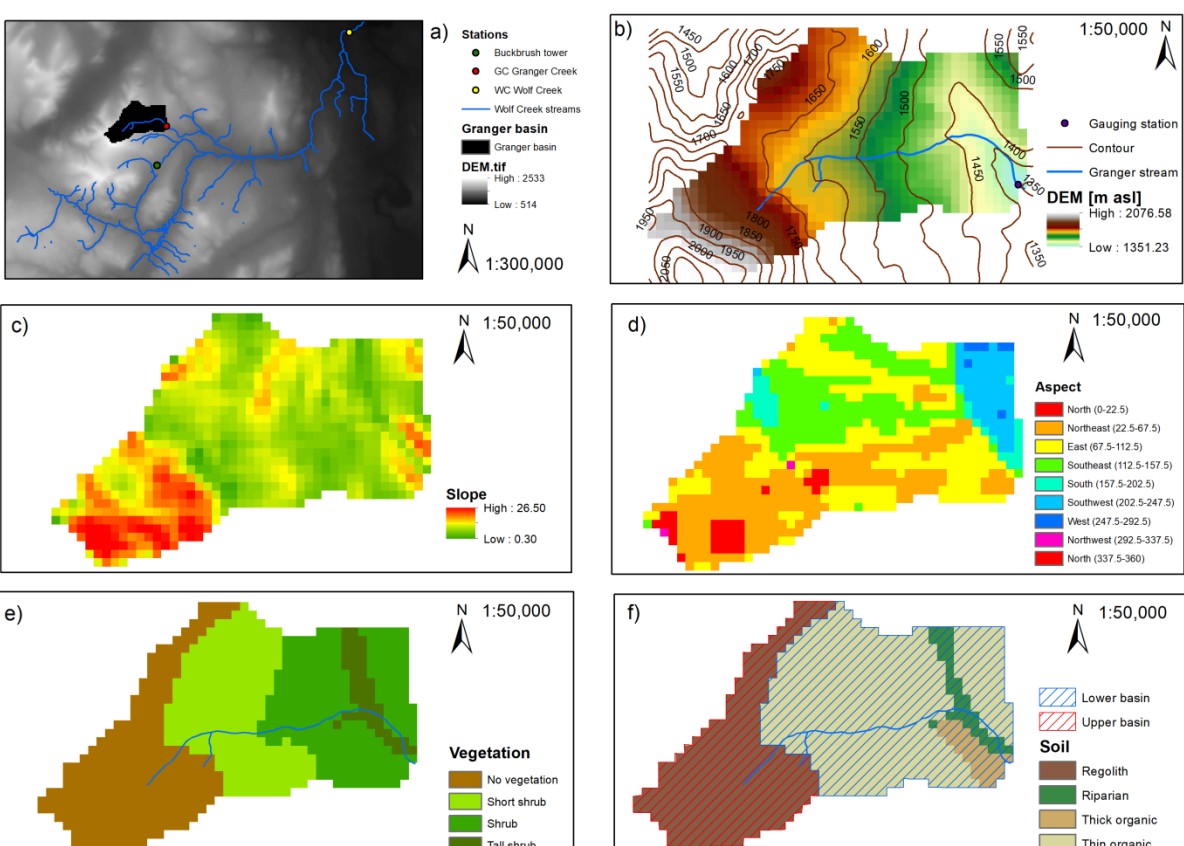

**Figure 1: Granger Basin characteristics showing a) location within the Wolf Creek catchment, stream network and gauging station, b) topography, c) slope, d) aspect, e) vegetation types and f) soil types. Lower (LB) and upper basin (UB) areas are marked.**



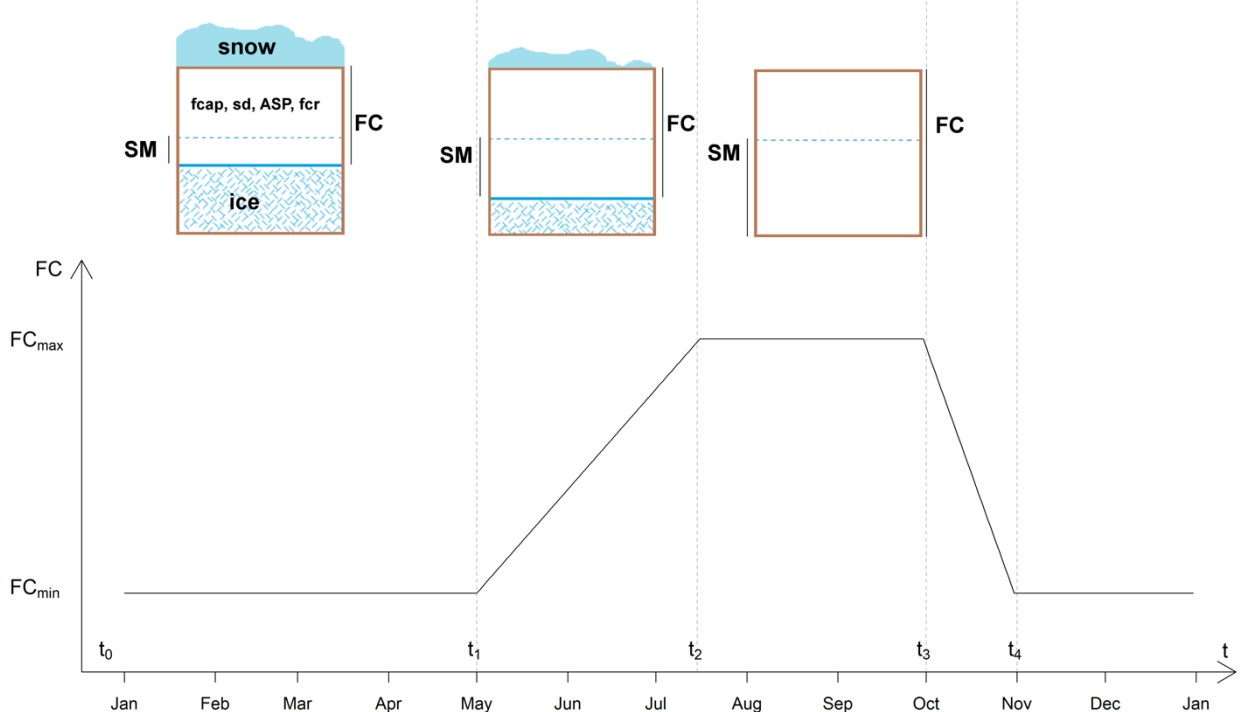

**Figure 2: Field capacity (*FC*) parameter set to be time-variable. *FC* is set to a minimum value (*FC_{min}*) when t<t_1 or t>t_4, t increasing linearly between t_1 and t_2 up to the maximum value *FC_{max}* and linearly decreasing between t_3 and t_4. The top image is a schematic representation of a soil cell to help visualize the variable *FC*. *FC* is dependent on the soil parameters: *fcap* (volumetric field capacity), *sd* (soil depth), *ASP* (aspect) and *fcr* (field capacity rate, calibrated parameter).**





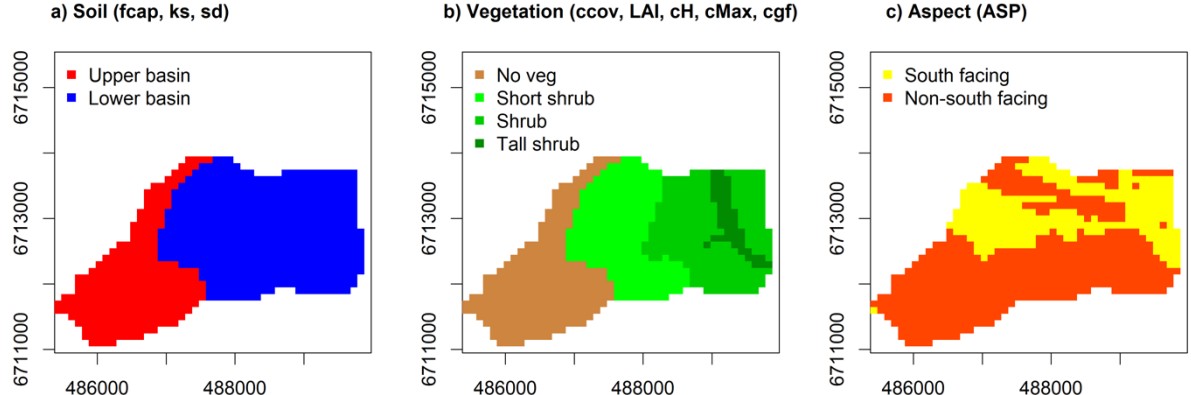

**Figure 3: Aggregation used to set the spatially variable parameters: a) soil classification for two calibrated parameters (fcap = volumetric field capacity, ks = recession coefficient) and one fixed parameter (sd = soil depth); b) vegetation classification for 5 fixed parameters (ccov=canopy coverage, LAI = leaf area index, cH = canopy height, cMax = maximum canopy storage and cgf = canopy gap fraction); c) aspect classification for ASP parameter that was calibrated if south-facing or set to 1 for any non-south-facing aspect.**





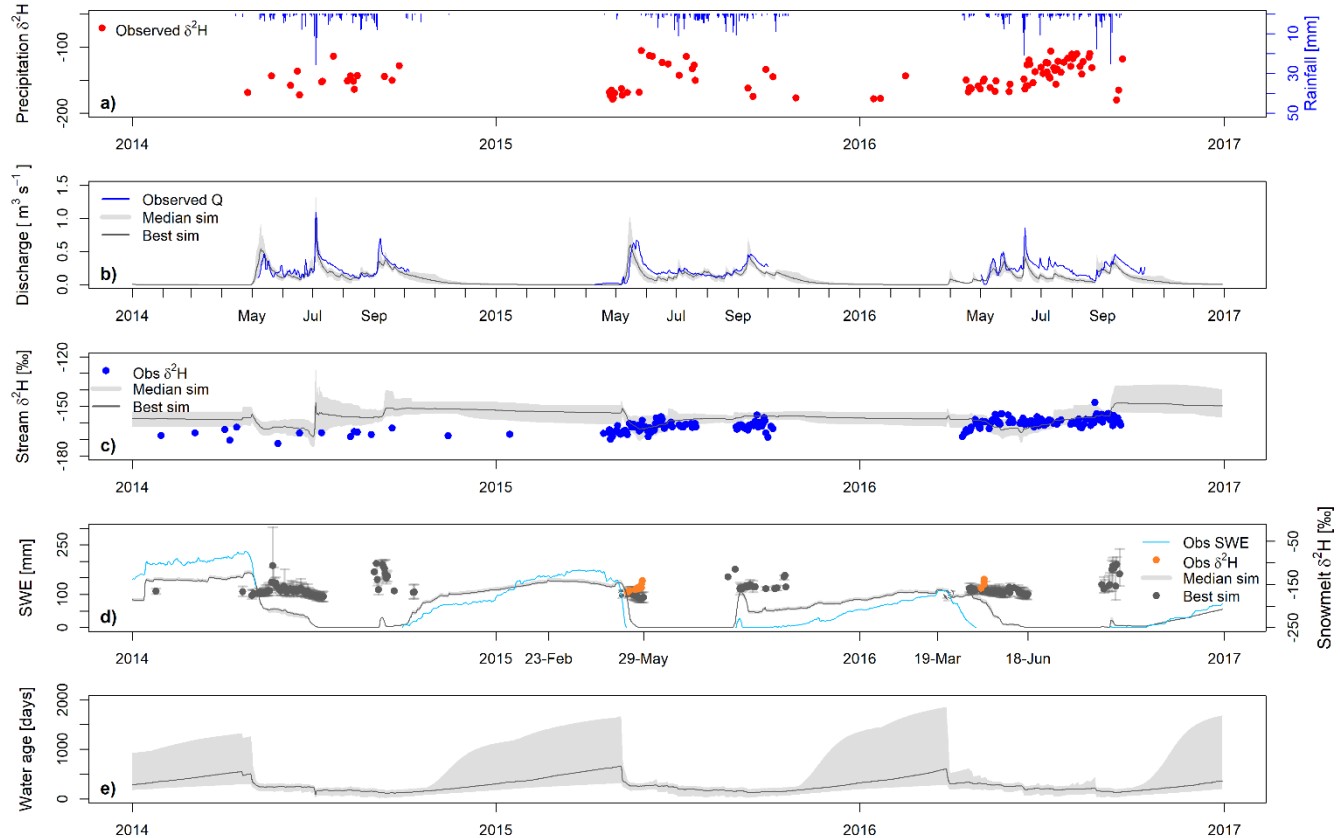

**Figure 4: Time series of a) observed precipitation isotopic signature of δ²H [‰] and observed rainfall [mm]. Time series of simulated (grey) and observed: b) discharge Q [m³s⁻¹]; c) stream water isotopic signature δ²H [‰]; d) catchment average snow water equivalent SWE [mm] and snowmelt isotopic signature δ²H [‰]; e) stream water ages [days] at outlet. Dark grey bands show ranges of selected best 100 runs ("Best sim"), black lines are the median simulation ("Median sim") among the 100 best simulations ("Best sim").**



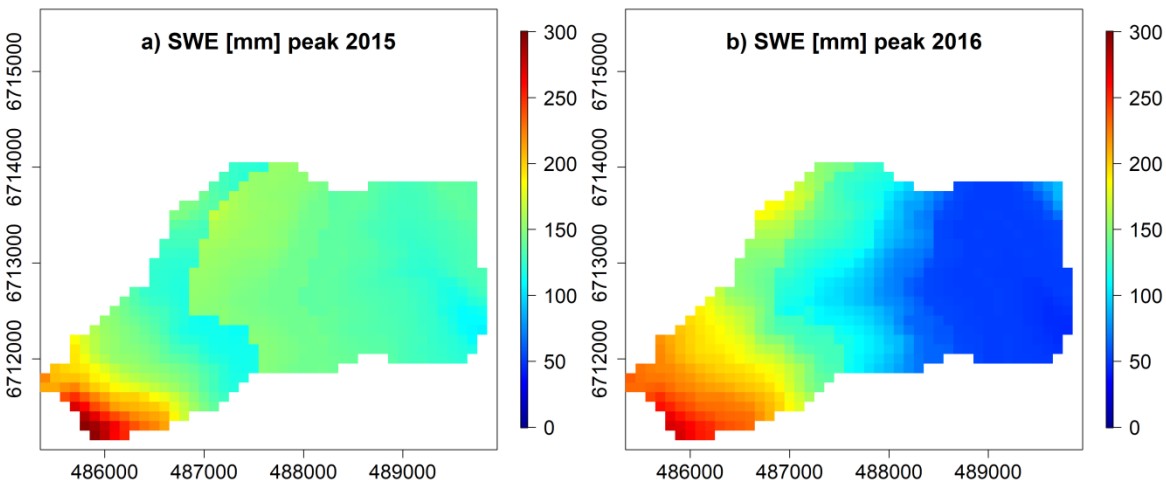

**Figure 5: Spatial distribution of snow water equivalent SWE [mm] for the day of peak SWE in both study ears. a) 23 February 2015 and b) 19 March 2016.**

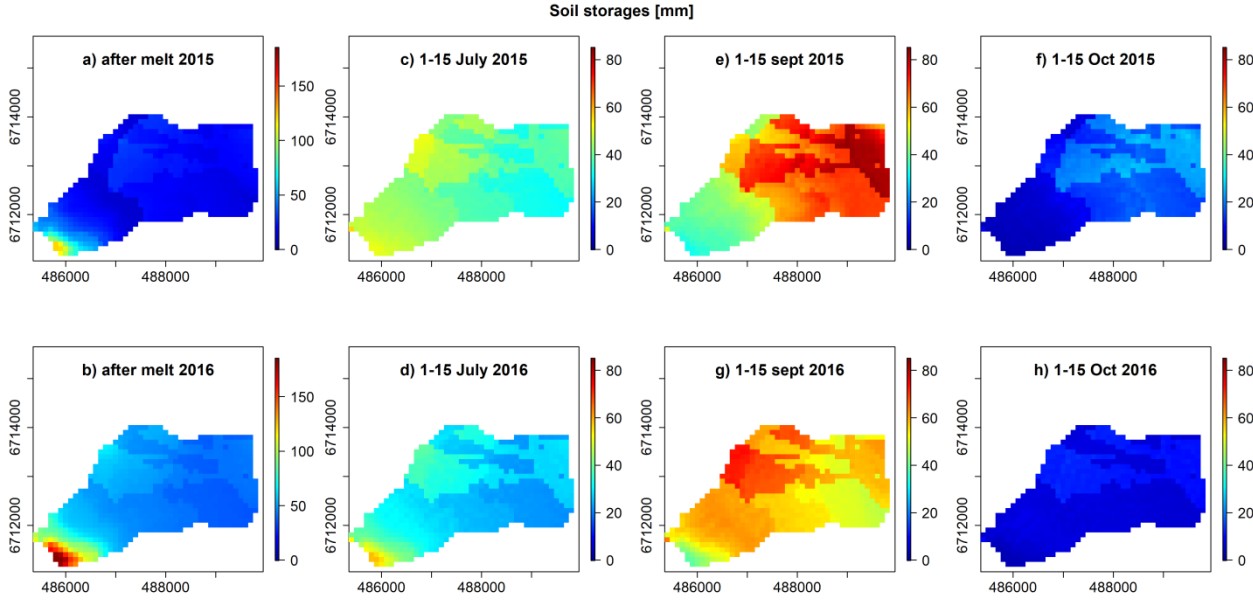

5    **Figure 6: Spatial distribution of simulated soil storages [mm]. Daily snapshots on first snow-free day in the year: (a) 28 May 2015 and b) 18 June 2016). Values averaged over 15 days: c) 1-15 July 2015, d) 1-15 July 2016, e) 1-15 September 2015, f) 1-15 September 2016, g) 1-15 October 2015 and h) 1-15 October 2016. Note scale in a) and b) are different to the scales in c) –h).**





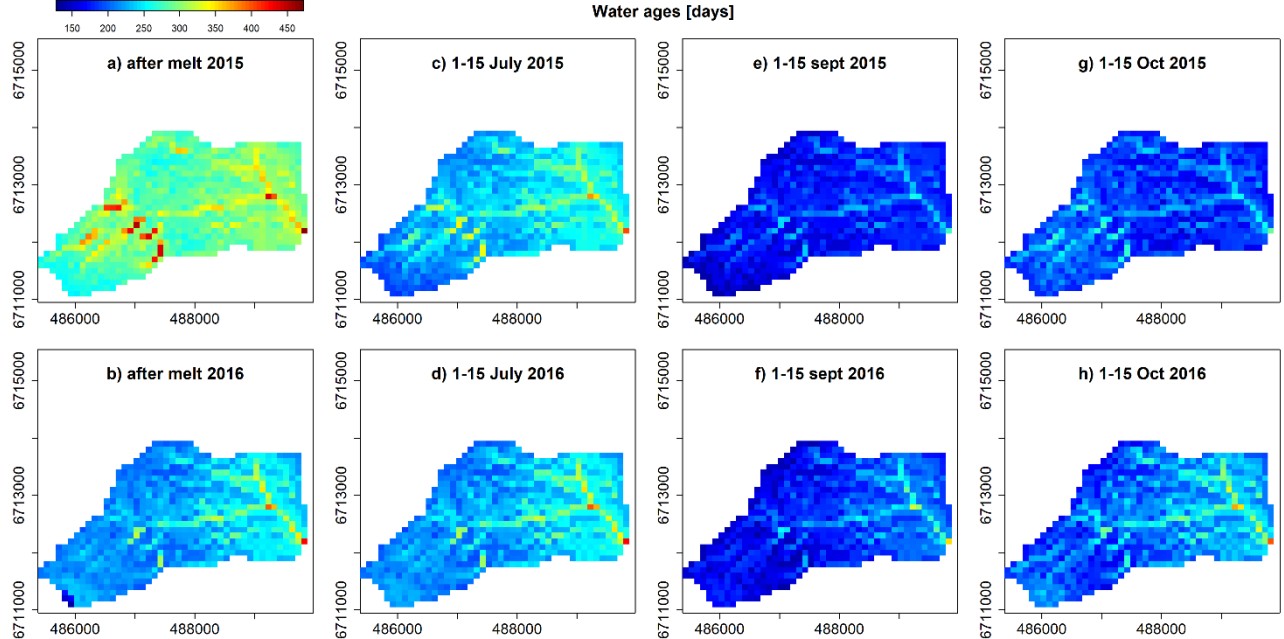

**Figure 7: Spatial distribution of simulated water ages [days] of runoff contributions (overland and lateral flow) from model cells. Daily snapshots on first snow-free day in the year: (a) 28 May 2015 and b) 18 June 2016). Values averaged over 15 days: c) 1-15 July 2015, d) 1-15 July 2016, e) 1-15 September 2015, f) 1-15 September 2016, g) 1-15 October 2015 and h) 1-15 October 2016.**