# Peer review of "Spatially-distributed tracer-aided runoff modelling and dynamics of storage and water ages in a permafrost-influenced catchment"

_Hydrology and Earth System Sciences, 2019_

## Referee Comment (RC1) · Anonymous Referee #1 · 14 Mar 2019

This study contributes to improving the understanding of runoff generation in an alpine discontinuous permafrost setting. It makes use of a conceptual model to simulate the seasonally changing stable isotope signature in the water of a small catchment. The STARR model used has been described in published literature, but the present paper applies the model in a spatially distributed fashion to study the 'age dynamics' of water storage and generated flow. The simulations show the displacement of old water from soil storage to the stream during snowmelt and the role of organic soils is explained. Figures 6 and 7 provide convenient summaries of the spatial distribution of soil storage and water age in the course of two years.

[Figure]

The Introduction section may be considered by some to be too lengthy but since many readers may not be familiar with the hydrological environment of the subarctic, such an extensive presentation can be helpful. I note that in Figure 4e, the range of the best 100 runs of stream water age has a considerably wider spread of values above than below the median. Is there an explanation for this asymmetry? The paper reads well, but some minor editorial changes are needed. p. 8, line 15: two open brackets in a sentence - in front of the word 'approximately' may be a comma? p. 10, line 22-23: I do not understand the phrase "as overland flow and only occasionally simulated during melt". p. 13, line 30: I do not understand "their high porosity allow several hundred mm of total water storage when saturated and large amounts at field capacity". p. 13, lines 34-35: "infiltrating meltwater supplies latent heat to bring soils to freezing". I thought the addition of latent heat would raise the soil temperature and not bring it down to freezing. p 14, line 4: "which are inked to conditions (of) the previous fall" – the word 'of' is missing.

---

## Referee Comment (RC2) · Anonymous Referee #2 · 19 Mar 2019

Major comments

The manuscript entitled "Spatially-distributed tracer-aided runoff modelling and dynamics of storage and water ages in a permafrost-influenced catchment" by Thea I. Piovano et al. developed a new permafrost feature that facilitates fully distributed simulations of hydrological storage dynamics and runoff processes, isotopic composition, and water ages within the Spatially distributed Tracer-Aided Rainfall-Runoff (STARR) conceptual model. The new feature is definitely very interesting to readers and a great advancement. One of the most important findings in this paper is that "Results from the model output correspond with previous field investigation and hydrograph separation studies

that indicate relatively old water (pre-event) dominates runoff generation during spring freshet." This result corresponds to findings by Suzuki et al. (2006b, 2018). This implied that further global warming might reduce permafrost coverage and speed up the hydrological cycle. Overall, authors need to revise the manuscript before its publication. Although there are some issues, I recommend that this paper be published after a few revisions are made.

First, there is a very important discrepancy in stream $\delta^2 H$ between the model simulation and observation data during snowmelt season. The $\delta^2 H$ in snow is low enough to be comparable with the $\delta^2 H$ in the stream; however, the trend in the latter is the complete opposite of the observations, because observed $\delta^2 H$ increases while the simulated $\delta^2 H$ decreased during the entire snowmelt season. I think that this is a critical flaw in the model because snowmelt water should primarily contribute at the beginning of the snowmelt season, when the surface soil is frozen. In a permafrost region, the active layer—which is a seasonal frost layer above the permafrost—strongly controls peak discharge (see, for instance, Yamazaki et al., 2006) and material transport (for instance, Suzuki et al., 2006a). Most researchers are interested in how seasonal active layer depth affects water age and isotope composition. I think that Suzuki et al. (2006a) showed that $\delta^{18} O$, which had a strong linear correlation with $\delta^2 H$ (for instance,Piovano et al., 2018), clearly increased during a snowmelt period. This suggested that the trend in a small Siberian basin would be similar with changes in the Granger basin. Thus, I believe that the new STARR feature has some problems in terms of isotope ratio estimation in permafrost influenced basins. Please add some discussion in terms of this aspect.

Second, I recommend that you emphasize how the permafrost and active layer affect water age and snowmelt runoff generation. To justify the role of old water in the permafrost regions, please consider previous studies in the Siberian watershed, such as Suzuki et al. (2006b), Yamazaki et al. (2006), and Suzuki et al. (2018).

Third, it would be better to add an additional comparison of water age during a

snowmelt from the previous study (Piovano et al., 2018) against the present study to evaluate the effects of permafrost with respect to the generation of snowmelt runoff. Otherwise you might discuss the effect of permafrost on water age using an additional experiment with and without seasonal changes in field capacity.

Fourth, I agree with your conclusion that "Results from the model output correspond with previous field investigation and hydrograph separation studies that indicate relatively old water (pre-event) dominates runoff generation during spring freshet. The relatively flashy nature of spring freshet in this largely frozen alpine catchment may seem counter-intuitive to this finding, yet water stored within the catchment from the previous year is the main source of stream water at the end of the melt season and explains isotopic damping of the signal." I think that this finding is coincident with Suzuki et al. (2018) in terms of continental-scale Arctic river basins. Thus, I recommend that you add how the role of permafrost in keeping water frozen during winter can mitigate the speeding up of the hydrological cycle (rainfall/snowfall to discharge).

Finally, please edit your text more carefully. For instance, please consider rewriting lines 17-21 on page 8 because those sentences are not clear. In addition, please add the word "liquid" to the figure 6 caption.

References: Piovano, T. I., Tetzlaff, D., Ala-aho, P., Buttle, J., Mitchell, C. P. J. and Soulsby, C.: Testing a spatially distributed tracer-aided runoff model in a snow-influenced catchment: Effects of multicriteria calibration on streamwater ages, Hydrol. Process., 32(20), 3089–3107, doi:10.1002/hyp.13238, 2018.

Suzuki, K., Konohira, E., Yamazaki, Y., Kubota, J., Ohata, T. and Vuglinsky, V.: Transport of organic carbon from the Mogot Experimental Watershed in the southern mountainous taiga of eastern Siberia, Nordic Hydrology, 37(3), 303–312, doi:10.2166/nh.2006.015, 2006a.

Suzuki, K., Kubota, J., Ohata, T. and Vuglinsky, V.: Influence of snow ablation and frozen ground on spring runoff generation in the Mogot Experimental Watershed, southern mountainous taiga of eastern Siberia, Nordic Hydrology, 37(1), 21–29, doi:10.2166/nh.2005.027, 2006b.

Suzuki, K., Matsuo, K., Yamazaki, D., Ichii, K., Iijima, Y., Papa, F., Yanagi, Y. and Hiyama, T.: Hydrological Variability and Changes in the Arctic Circumpolar Tundra and the Three Largest Pan-Arctic River Basins from 2002 to 2016, Remote Sensing, 10, 402–, doi:10.3390/rs10030402, 2018.

Yamazaki, Y., Kubota, J., Ohata, T., Vuglinsky, V. and Mizuyama, T.: Seasonal changes in runoff characteristics on a permafrost watershed in the southern mountainous region of eastern Siberia, Hydrol. Process., 20(3), 453–467, doi:10.1002/hyp.5914, 2006. My main concerns are as follows:

Please also note the supplement to this comment:
https://www.hydrol-earth-syst-sci-discuss.net/hess-2019-59/hess-2019-59-RC2-supplement.pdf

---

## Author Comment (AC1) · 29 Apr 2019

**Response Referee 1**

This study contributes to improving the understanding of runoff generation in an alpine discontinuous permafrost setting. It makes use of a conceptual model to simulate the seasonally changing stable isotope signature in the water of a small catchment. The STARR model used has been described in published literature, but the present paper applies the model in a spatially distributed fashion to study the "age dynamics" of water storage and generated flow. The simulations show the displacement of old water from soil storage to the stream during snowmelt and the role of organic soils is explained.

[Figure]

Figures 6 and 7 provide convenient summaries of the spatial distribution of soil storage and water age in the course of two years.
*Response: Thank you for this positive evaluation of our study.*

The Introduction section may be considered by some to be too lengthy but since many readers may not be familiar with the hydrological environment of the subarctic, such an extensive presentation can be helpful. I note that in Figure 4e, the range of the best 100 runs of stream water age has a considerably wider spread of values above than below the median. Is there an explanation for this asymmetry?
*Response: The distribution of stream water ages is asymmetrical as only a few of the best simulation show higher ages. Such asymmetry in the spread of modelled ranges has been already shown in some of the previous model applications (e.g. Ala-aho et al., 2017b; Piovano et al., 2018). In the latter it was noted this asymmetry was controlled by the parameter ranges used in the multi-criteria calibration. Here, a key factor for the increase in the uncertainty during winter periods is the lack of constraints in that specific period (i.e. during winter both data of discharge and isotope stream signature were not available). In the revised version we have plotted the stream water ages in a semi-log scale to help visualizing the trend when uncertainties were smaller.*

The paper reads well, but some minor editorial changes are needed.
*Response: Thanks for spotting these errors and inaccuracies.*

p. 8, line 15: two open brackets in a sentence - in front of the word "approximately" may be a comma?
*Response: Done.*

p. 10, line 22-23: I do not understand the phrase "as overland flow and only occasionally simulated during melt".
*Response: Corrected.*

p. 13, line 30: I do not understand "their high porosity allow several hundred mm of total water storage when saturated and large amounts at field capacity".

*Response: Corrected.*

p. 13, lines 34-35: "infiltrating meltwater supplies latent heat to bring soils to freezing". I thought the addition of latent heat would raise the soil temperature and not bring it down to freezing.
*Response: You are correct. We changed the text.*

p 14, line 4: "which are inked to conditions (of) the previous fall" – the word "of" is missing.
*Response: Added.*

---

## Author Comment (AC2) · 29 Apr 2019

**Response Referee 2**

The manuscript entitled "Spatially-distributed tracer-aided runoff modelling and dynamics of storage and water ages in a permafrost-influenced catchment" by Thea I. Piovano et al. developed a new permafrost feature that facilitates fully distributed simulations of hydrological storage dynamics and runoff processes, isotopic composition, and water ages within the Spatially distributed Tracer-Aided Rainfall Runoff (STARR) conceptual model. The new feature is definitely very interesting to readers and a great advancement. One of the most important findings in this paper is that "Results from the model

output correspond with previous field investigation and hydrograph separation studies that indicate relatively old water (pre-event) dominates runoff generation during spring freshet." This result corresponds to findings by Suzuki et al. (2006b, 2018). This implied that further global warming might reduce permafrost coverage and speed up the hydrological cycle. Overall, authors need to revise the manuscript before its publication. Although there are some issues, I recommend that this paper be published after a few revisions are made.

My main concerns are as follows: (1) First, there is a very important discrepancy in stream $\delta^2$H between the model simulation and observation data during snowmelt season. The $\delta^2$H in snow is low enough to be comparable with the $\delta^2$H in the stream; however, the trend in the latter is the complete opposite of the observations, because observed $\delta^2$H increases while the simulated $\delta^2$H decreased during the entire snowmelt season. I think that this is a critical flaw in the model because snowmelt water should primarily contribute at the beginning of the snowmelt season, when the surface soil is frozen. In a permafrost region, the active layer - which is a seasonal frost layer above the permafrost - strongly controls peak discharge (see, for instance, Yamazaki et al., 2006) and material transport (for instance, Suzuki et al., 2006a). Most researchers are interested in how seasonal active layer depth affects water age and isotope composition. I think that Suzuki et al. (2006a) showed that $\delta^{18}$O, which had a strong linear correlation with $\delta^2$H (for instance, Piovano et al., 2018), clearly increased during a snowmelt period. This suggested that the trend in a small Siberian basin would be similar with changes in the Granger basin. Thus, I believe that the new STARR feature has some problems in terms of isotope ratio estimation in permafrost influenced basins. Please add some discussion in this aspect.

*Response: With respect we are not sure what we are being asked to do here. In the model for 2014 and 2015, a slight decline in isotopes is predicted. This over-estimates measured values in 2014, but whilst values are simulated OK in 2015, the depression wasn't measured. We wouldn't agree that the simulations were "the complete opposite" of observations. As we try and explain in the discussion, the data suggest that in the*

*early melt some of the soil isn't frozen (likely in south-facing, low altitude areas) which allows mixing and damping of the melt signal. We now refer to the Suzuki (2016a) paper; although this has large gaps in the isotope data, but it implies a slight depression in the early snowmelt, then an increase afterwards, so it isn't entirely comparable.*

(2) Second, I recommend that you emphasize how the permafrost and active layer affect water age and snowmelt runoff generation. To justify the role of old water in the permafrost regions, please consider previous studies in the Siberian watershed, such as Suzuki et al. (2006b), Yamazaki et al. (2006), and Suzuki et al. (2018).
*Response: With respect, we do think that section 5.2 extensively discusses the mechanisms for the mobilization of the old water in this environment and provides a comparison with results presented in other work relevant to our context. Although we think that the main focus, approaches and conclusions of the suggested references are not entirely consistent with the work presented, we now cite some of the papers listed where appropriate.*

(3) Third, it would be better to add an additional comparison of water age during a snowmelt from the previous study (Piovano et al., 2018) against the present study to evaluate the effects of permafrost with respect to the generation of snowmelt runoff. Otherwise you might discuss the effect of permafrost on water age using an additional experiment with and without seasonal changes in field capacity.
*Response: We have now added more emphasis on the comparison with the previous study in the discussion. Previous studies, in snow-dominated catchments showed a predominance of young water during melt. So, this highlights the different effects of the permafrost and thaw on water ages.*

(4) Fourth, I agree with your conclusion that "Results from the model output correspond with previous field investigation and hydrograph separation studies that indicate relatively old water (pre-event) dominates runoff generation during spring freshet. The relatively flashy nature of spring freshet in this largely frozen alpine catchment may seem counter-intuitive to this finding, yet water stored within the catchment from the

previous year is the main source of stream water at the end of the melt season and explains isotopic damping of the signal." I think that this finding is coincident with Suzuki et al. (2018) in terms of continental-scale Arctic river basins. Thus, I recommend that you add how the role of permafrost in keeping water frozen during winter can mitigate the speeding up of the hydrological cycle (rainfall/snowfall to discharge).

*Response: Thank you for these suggestions. We expanded the discussion in order to mention this important implication and refer to this recent paper.*

Finally, please edit your text more carefully. For instance, please consider rewriting lines 17-21 on page 8 because those sentences are not clear. In addition, please add the word "liquid" to the figure 6 caption.

*Response: Thanks for your suggestion. We carefully proof read the entire revised manuscript and rephrased some sentences.*

Reference: Piovano, T. I., Tetzlaff, D., Ala-aho, P., Buttle, J., Mitchell, C. P. J. and Soulsby, C.: Testing a spatially distributed tracer-aided runoff model in a snow-influenced catchment: Effects of multicriteria calibration on streamwater ages, Hydrol. Process., 32(20), 3089–3107, doi:10.1002/hyp.13238, 2018.

Suzuki, K., Konohira, E., Yamazaki, Y., Kubota, J., Ohata, T. and Vuglinsky, V.: Transport of organic carbon from the Mogot Experimental Watershed in the southern mountainous taiga of eastern Siberia, Nordic Hydrology, 37(3), 303–312, doi:10.2166/nh.2006.015, 2006a.

Suzuki, K., Kubota, J., Ohata, T. and Vuglinsky, V.: Influence of snow ablation and frozen ground on spring runoff generation in the Mogot Experimental Watershed, southern mountainous taiga of eastern Siberia, Nordic Hydrology, 37(1), 21–29, doi:10.2166/nh.2005.027, 2006b.

Suzuki, K., Matsuo, K., Yamazaki, D., Ichii, K., Iijima, Y., Papa, F., Yanagi, Y. and Hiyama, T.: Hydrological Variability and Changes in the Arctic Circumpolar Tundra and the Three Largest Pan-Arctic River Basins from 2002 to 2016, Remote Sensing, 10, 402–, doi:10.3390/rs10030402, 2018.

[Figure]

Yamazaki, Y., Kubota, J., Ohata, T., Vuglinsky, V. and Mizuyama, T.: Seasonal changes in runoff characteristics on a permafrost watershed in the southern mountainous region of eastern Siberia, Hydrol. Process., 20(3), 453–467, doi:10.10
* * *

---

## Author Response (AR2)

We thank the editor for the positive comment and for suggesting the typographic and presentation corrections. We have carefully correct them and in the following document we are providing the marked-up revised version.

[revised manuscript text omitted]